# Analysis of Same Selected Immunomodulatory Properties of Chorionic Mesenchymal Stem Cells

**Darina Bačenková** [1,*] , **Marianna Trebuňová** [1] , **Lukáš Zachar** [2] , **Radovan Hudák** [1] , **Gabriela Ižaríková** [3] , **Katarína Šurínová** [1] **and Jozef Živčák** [1]

1 Department of Biomedical Engineering and Measurement, Faculty of Mechanical Engineering, Technical University of Košice, Letná 9, 042 00 Košice, Slovakia; marianna.trebunova@tuke.sk (M.T.); radovan.hudak@tuke.sk (R.H.); sromovska.katarina@gmail.com (K.Š.); jozef.zivcak@tuke.sk (J.Ž.)
2 Department of Laboratory Medicine, University Hospital of Louis Pasteur, Trieda SNP 1, 041 66 Košice, Slovakia; zachar0lukas@gmail.com
3 Department of Applied Mathematics and Informatics, Faculty of Mechanical Engineering, Technical University of Košice, Letná 9, 042 00 Košice, Slovakia; gabriela.izarikova@tuke.sk
* Correspondence: darina.bacenkova@tuke.sk; Tel.: +42-1055-602-2380

**Abstract:** Mesenchymal stem cells (MSCs) represent a population of adherent cells that can be isolated from multiple adult tissues. MSCs have immunomodulatory capacity and the ability to differentiate into many cell lines. Research study examines the immunomodulatory properties of MSCs isolated from chorion (CMSCs). Following the stimulation process, it was found that MSCs are capable of immunomodulatory action via the release of bioactive molecules as well as through direct contact with the immune cells. Immunomodulatory potential of the CMSCs was analyzed by modifying proliferative capacity of mitogen-activated lymphocytes. CMSCs and lymphocytes were tested in cell-to-cell contact. Lymphocytes were stained with carboxyfluorescein diacetate succinimidyl ester. Inhibition of the proliferation of activated lymphocytes was observed. Following the co-cultivation, the expression of markers involved in the immune response modulation was assessed. Afterwards, an increase in CMSCs expression of IL-10 was detected. Following the co-cultivation with activated lymphocyte, adhesion molecules CD54 and CD44 in the CMSCs increased. An increase of CD54 expression was observed. The properties of CMSCs, adherence and differentiation ability, were confirmed. The phenotype of CMSCs CD105+, CD90+, CD73+, CD44+, CD29+, CD45−, CD34−, CD54+ was characterized. It was demonstrated that chorion-derived MSCs have important immunomodulatory effects.

**Keywords:** chorionic mesenchymal stem cells; immunomodulation; lymphocytes

## 1. Introduction

Mesenchymal stem cells (MSCs) form a population of adherent cells with a characteristic phenotype in the absence of hematopoietic markers [1]. MSCs were first described by Friedenstein as fibroblast-like cells in the bone marrow [2]. In addition to bone marrow, adult MSCs were also isolated from adipose tissue, liver and muscle tissue [3]. A rich source of stem cells is present in placenta, fetal packs, Wharton's gel, umbilical cord blood and amniotic fluid [4–8]. The fetus and protective fetal membranes have properties of semi-allogenic tissue related to the uterus in the mother's body [9].

In vitro, MSCs are characterized by their adherence to a plastic surface, fibroblast-like bipolar shaped cells morphology and a broadly defined phenotype. The expression of typical surface features for the MSCs population is as follows: CD105+, CD90+, CD73+, CD45−, CD34−, HLA-DR-. In addition, the MSCs have the following properties: CD9+, CD29+, CD44+, CD49+, CD54+, CD61+, CD63+,

CD71+, CD97+, CD98+, CD99+, CD106+, CD146+, CD155+, Stro-1+, CD166+, CD166+, CD271+, CD276+ and CD304+. These cells are characterized by the absence of hematopoietic markers, such as CD34−, CD79−, CD19− and major histocompatibility complex (MHC) class II molecules [10,11].

Mesenchymal stem cells are involved in multiple interactions in the body. They act as immunomodulatory, antifibrosis and angiogenes [12]. These cells also have the ability to create an immunomodulatory microenvironment and thus they help to minimize organ damage caused by inflammation and cells activated by the immune system [13]. The characteristic feature of stem cells is their ability to self-revise and differentiate into several types of tissues, predominantly mesenchymal as well as endo- and ectodermal tissues types. Numerous studies have described their differentiation into osteoblasts, chondrocytes, adipocytes, myocytes, epithelial cells and cardiomyocytes. Likewise, MSCs are capable of differentiating into neuroectodermal cells: neurons, astrocytes, oligodendrocytes and endodermal cells: hepatocytes [14–19].

The immunomodulatory properties of MSCs are directly modulated by the environment where they are located. MSCs are modulated by various paracrine factors occurring in the inflammatory environment. The inflammatory environment also affects the cells of the adaptive immune system, especially T lymphocytes. MSCs were found to interact closely with immune cells, such as lymphocytes [20]. MSCs activated in this environment produce the paracrine-acting factors that act on cells of the immune system [21]. MSCs are also characterized by their mobility, and their ability to migrate to a higher incidence of inflammatory cytokines [22]. Following adequate stimulation, they are able to act by multifactor, direct contact and secretion of bioactive molecules, transforming growth factor β1 (TGF-β1), indoleamine 2,3-dioxygenase (IDO), hepatocyte growth factor (HGF), prostaglandin E2 (PGE2) and interleukin (IL-10) [23].

There are several causes of inflammation throughout the process, such as human leukocyte antigen ABC(HLA-ABC), CD90, activated leukocyte adhesion molecule CD166 (ALCAM) and other integrins that allow interactions of MSCs and T lymphocytes. Repeatedly, the inhibition of the proliferation of activated lymphocytes was observed during the co-cultivation of MSCs and mitogen-induced T lymphocytes [24]. At higher concentrations (about 10–40 MSCs per 100 stimulated lymphocytes), MSCs are able to inhibit the proliferation of T lymphocytes. On the contrary, lower concentrations of MSCs (0.1–1%) can stimulate lymphocyte proliferation in mixed lymphocyte culture [25]. An accurate understanding of how reactions communicate with the interacting population of MSCs and lymphocytes is important for the therapeutic use of MSCs as well as for the optimization of the applied dose of cells. Immunomodulatory properties as well as immunosuppressive effect of MSCs were used multiple times in patients with graft versus host disease (HVGR), after allogeneic bone marrow transplantation [26,27]. Recent studies show that several types of immune cells affect the immunomodulatory function of MSCs. The detailed course of these interactions remains unclear. Several mechanisms of interaction of MSCs and immune cells were described. In the intercellular contact of MSCs and immune system cells, activation secretion is triggered and in some cases immune cell apoptosis can be induced.

Individual subpopulations of T regulatory (Treg) lymphocytes are characterized by specific phenotypes. Treg lymphocytes have regulatory properties and an ability to control the immune response. Mechanisms involved in the development of the regulatory function of Treg lymphocytes are more complex and require multiple signals [28]. CD4+, CD25+, and FoxP3+ Treg lymphocytes play an important role in the control of immunological tolerance and homeostasis of the immune system [29]. MSCs not only have the ability to influence the proliferation of activated lymphocytes, but they can also differentiate them. They preferably act on the differentiation and expansion of T lymphocytes and their subgroups with regulatory phenotype [30]. They are involved in modulating the subpopulation of CD4+ and CD8+ T lymphocytes, resulting in their differentiation into Treg lymphocytes which are known for their inhibitory action on activated lymphocytes [31,32]. CD4+, CD25+, and FoxP3+ Treg lymphocytes suppress autoimmune reactions and produce IL-10, TGF-β cytokines. They act primarily through direct contact. MSCs are involved in coordinating the function of immune cells in a particular environment. Immune system and cells associated with the immune system are linked by complex

processes of expression and production of cytokines. The topic of the study of the biological and immunomodulatory properties of MSCs isolated from fetal packaging is highly relevant. So far, all the developments and interactions in this area have not been explored.

## 2. Materials and Methods

### 2.1. Ethical Guidelines

The study was conducted in accordance with the Declaration of Helsinki (1964) and approved by the local ethics committee of the Louis Pasteur University Hospital in Košice (Slovakia). The study number was 2017/EK/1027, the project code was APVV-17-0118, and the date of approval was 30 November 2017. All samples were obtained from healthy donors after they gave their informed written consent.

### 2.2. Isolation and Culture of Human Chorionic Mesenchymal Stem Cells

Human chorionic mesenchymal stem cells (CMSCs) were isolated from the part of the extraembryonic membranes, chorion. Human term placentas by section (38–40-week gestation) were obtained from healthy donor mothers. Further use of placental tissues was performed with the donor's agreement. The amnion and chorion were then manually separated and chorion was washed in phoshate-buffered saline (PBS) (Thermo Fisher Scientific, New York, NY, USA) containing penicillin 100 IU/mL, streptomycin 100 µg and amphotericin 0.25 µg (Thermo Fisher Scientific). After isolating a $10 \times 10$ cm part of chorion from amnion, the tissue was washed and cut into small pieces of $0.5 \times 0.5$ cm. Tissue was digested of 15 min at 37 °C in dispase II 2.4 U/mL (Sigma-Aldrich, St. Louis, MO, USA) in Dulbecco's Modified Eagle's Medium (DMEM) (Thermo Fisher Scientific). Small pieces were washed and then centrifuged at 150 g-force for 15 min. Chorion pieces were incubated for approximately 90 min at 37 °C in DMEM (Thermo Fisher Scientific) containing 1 mg/mL collagenase type II (Thermo Fisher Scientific). Cells were collected by centrifugation at 150 g for 15 min. Dispersed chorion cells were filtered through a sterile Falcon® cell strainer 40 µm (Thermo Fisher Scientific). Chorionic cells were cultured in vitro at a density of 4000 cells/cm$^2$ in alpha Minimal Essential Medium (alpha MEM) GlutaMAX supplement (Thermo Fisher Scientific) supplemented with 10% fetal calf serum (FCS) (Thermo Fisher Scientific) and antibiotic-antimycotic solution (ATB) contained penicillin 100 IU/mL, streptomycin 100 µg and amphotericin 0.25 µg (Thermo Fisher Scientific). Isolated chorionic mesenchymal cells, called CMSCs were maintained at 37 °C in a humidified atmosphere with a 5% $CO_2$ atmosphere. During the first week all the medium and no adherent cells were removed and replaced with fresh medium. When the cells were more than 80% confluent, they were recovered with 0.25% trypsin/ethylene diamine tetra acetic acid (EDTA) (Thermo Fisher Scientific) and detached by trypsinization. The cells were released and trypsin ETDA was neutralized with 10% FCS. In the next passages, the cell suspension was used for testing immunomodulatory properties and phenotype characterization cells.

### 2.3. Clonogenic Potential CMSCs

Colony–Forming unit fibroblast (CFU-F) assays were performed on freshly isolated CMSCs. The cells were suspended at a final density of 1000 cells per cm$^2$ in Eagles' minimal essential medium (MEM) (Thermo Fisher Scientific) supplemented with 10% fetal bovine serum (FBS) (Thermo Fisher Scientific) and antibiotic-antimycotic solution containing penicillin 100 IU/mL, streptomycin 100 µg and amphotericin 0.25 µg in 6-well plate (Sarstedt, Nümbrecht, Germany). Cultures were incubated in a humidified atmosphere with 5% $CO_2$ at 37 °C. The cultures were ended on day 10. Romanowsky–Giemsa (Sigma-Aldrich) staining was used to determine clonogenicity and monitor the fibroblast colony formation.

### 2.4. Flow Cytometry Fenotyping CMSCs

Flow cytometry (FACS) was used to evaluate the expression of cell markers after the first passages. For the purpose of evaluation of surface marker expression, cell suspensions CMSCs of $5.0 \times 10^5$ cell/mL, volume of 100 µL PBS were incubated for 30 min with fluorochrome, phycoerythrin (PE) with conjugated antibodies CD105 PE clone 43A3 (BioLegend, San Diego, CA, USA), CD44 PE (Miltenyi Biotec, Bergisch Gladbach, Germany) clone G44-26 a CD90 PE clone 5E10 (BD Biosciences, Franklin Lakes, NJ, USA), CD73 PE clone AD2 (Miltenyi Biotec), CD29 PE (Miltenyi Biotec) a fluorescein isothiocyanate (FITC)—conjugated antibodies against human antigens CD146 FITC (Miltenyi Biotec), CD45 FITC clone HI30 (Pharmingen, USA), CD34 FITC clone 581 (BD Biosciences), Toll-like receptor (TLR)4 FITC clone MTS510 (Invitrogen, Thermo Fisher Scientific), CD54 PE intercellular adhesion molecule (ICAM-1) clone REA266 (Miltenyi Biotec), and HLA DR FITC clone AC122 (Miltenyi Biotec). Intracellular markers were identified in suspension with a concentration of $5.0 \times 10^5$ cells/mL. The cells were fixed in suspension and then permeabilized with 150 µL BD Cytoxix/Cytoperm (BD Biosciences). The cells suspension was incubated for 20 min at 4 °C. Afterwards, the sample was washed by BD Perm/Wash buffer (BD Biosciences). Single-cell suspensions were stained with fluorochrome labeled antibodies IL-10 PE Clone JES3-19F1 (BD Biosciences) and TLR3 PE clone TLR3.7 (Thermo Fisher Scientific). Cells samples were analyzed with a fluorescent activated cell sorter (FACS) Calibur (BD Biosciences). Collected data were analyzed and evaluated using BD CellQuest software (BD Biosciences) and FCS Express 5 (De Novo Software, Glendale, CA, USA).

### 2.5. Isolation and Stimulation of Lymphocytes

A fraction of mononuclear cells was extracted from whole blood using Ficoll-Paque (GE Healthcare, Chicago, IL, USA) and gradient centrifugation. Venous blood samples were used from seven donors in the experiment. Mononuclear cells were cultivated in vitro in RPMI 1640 medium (Thermo Fisher Scientific) with 10% FCS at 37 °C, and a 5% $CO_2$ atmosphere. Lymphocytes were stimulated by phytohemagglutinin (PHA) 5 µg/mL (Thermo Fisher Scientific). For the purpose of lymphocyte proliferation studies, fluorescent dyes carboxyfluorescein diacetate succinimidyl ester (CFDA-SE) 10 µM were used (Sigma-Aldrich, USA). Vital stains carboxylflurescenin succinidyl ester (CFSE) were used in co-cultivation lymphocyte with CMSCs. CFSE for the duration of two hours and examined by fluorescent microscopy and flow cytometer at wave length of 488 nm. CFSE spontaneously and irreversibly coupled to cellular proteins by reaction with lysine side chains and other available amines. After an initial period of equilibrium, the fluorescence of resting cells labeled with CFSE is stable over periods of months [33].

### 2.6. FACS Fenotyping of Lymphocytes

The phenotype of non-adherent PHA-stimulated CFSE lymphocytes after co-culture with CMSCs was determined by flow cytometry. For evaluation of surface marker expression, cell suspensions lymphocyte of $5.0 \times 10^5$ cell/mL, volume of 100 µL PBS were incubated for 30 min with fluorochrome—conjugated antibodies CD3 FITC clone SK7 (BD Biosciences), CD4 FITC clone M-T466 (Miltenyi Biotec), CD193 allophycocyanin (APC) clone 5E8.4, CD69 PE clone FN50 (BD Biosciences), CD119 FITC clone REA161 (Miltenyi Biotec) and the results were analyzed using flow cytometry.

### 2.7. Stimulation CMSCs with Poly (I:C)

We used activated cells to test the immunomodulatory properties of CMSCs. CMSCs were activated with polyinosine: polycytidylic acid (*Poly (I:C)*) (Thermo Fisher Scientific) with a 20 µg/mL synthetic analog of double-stranded RNA viruses and TLR3 ligand. *Poly (I:C)* was treated on a CMSC for about 48 h and MSCs after the passage were seeded into adherent culture flasks T-25 cm². Activated CMSCs *Poly (I:C)* were used for contact co-culture with activated CFSE-SE labeled lymphocytes.

### 2.8. Co-Cultivation CMSC and Lymphocytes

We evaluated in vitro the immunomodulatory properties of CMSCs at passage 2 and co-cultivated with PHA lymphocyte. We observed an effect on proliferation in three experimental groups of cells, groups of PHA lymphocytes and co-cultured with CMSCs *Poly (I:C)*, PHA lymphocytes with CMSCs without stimulation and PHA lymphocytes alone were compared. First, CMSCs *Poly (I:C)* cells or CMSCs were seeded into adherent culture flasks T-25. After 24 h we added PHA lymphocytes in proportion (1:4). CMSCs and lymphocytes were co-cultured 4 days in the culture medium Alpha MEM: RPMI 1640 (1:1) (Thermo Fisher Scientific) containing 10% FCS and 1% ATB at 37 °C in vitro in a 5% $CO_2$ atmosphere. Throughout the change in fluorescence intensity, we observed the degree of the effect of proliferation in lymphocytes CFSE co-cultivated with CMSCs. The impact rate of proliferation was evaluated after 4 days of co-cultivation. CFSE-SE labeled lymphocytes were analyzed using flow cytometry.

### 2.9. Statistical Analysis

Statistical comparison was performed using the Kruskal–Wallis test for paired samples with the software Statistica 9 (TIBCO Software, Palo Alto, CA, USA). Data were collected from seven independent experiments. This difference was statistically significant ($p < 0.05$).

## 3. Results

The main objective of the experimental work was to observe the immunomodulatory properties of CMSCs. We tested the effect of activated CMSCs on lymphocytes proliferation. We also focused on the detailed characterization of morphology and the determination of the CMSCs phenotype.

### 3.1. Isolation of CMSCs

CMSCs cells were isolated from seven placental samples of the mesenchymal layer from the chorionic membrane. Adherence of isolated CMSCs was recorded within 48 h following the isolation and in vitro culture of CMSCs in complete culture medium alpha MEM whit 10% FCS and ATB. In the first change of culture medium, non-adherent hematopoietic cells, erythrocytes and damaged cells were removed. These were admixed only in the initial stage of culture. After one week of culture of the CMSC, we observed the growth of fibroblast cells, characterized by an elongated shape. After passage 1, the culture of CMSCs was mostly homogeneous. Morphologically, the cells were spindle-shaped and adhered (see Figure 1a,b). An average ($11 \times 10^6$) CMSCs from seven chorionic membrane samples were isolated. A total of seven fetal membranes were processed; the population of adherent and morphologically fibroblast-like CMSCs cells were isolated from the chorionic membrane.

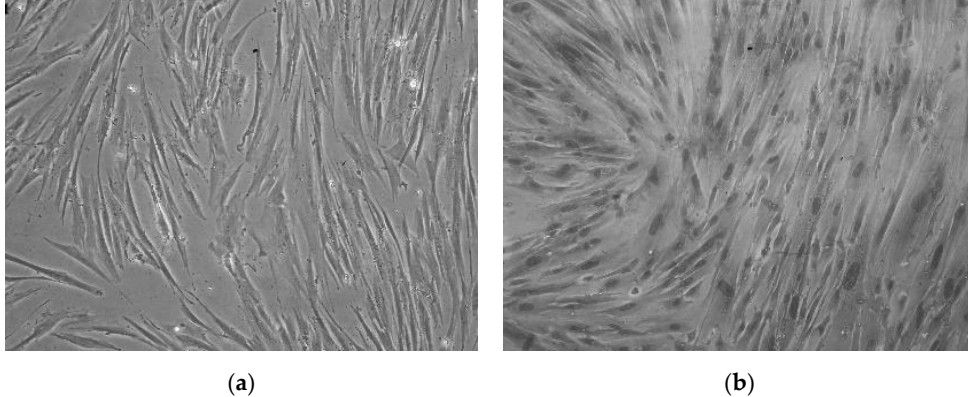

(**a**)                           (**b**)

**Figure 1.** (**a**) Monolayer culture human chorionic mesenchymal stem cells (CMSCs) after 10 days. (**b**) CMSCs characterized by an elongated shape; morphologically the cells are spindle-shaped and adherent (20× magnification of objective).

## 3.2. Clonogenic Potential CMSCs

In order to analyze clonogenetic potential, colony-forming unit fibroblast CFU-F assays were performed using CMSCs at passage number one. After 10 days of CMSCs cultivation at 37 °C and 5% $CO_2$ concentration, CFU-F formation was observed. CMSCs formed a population of cells with fibroblast cell morphology and high proliferation potential (see Figure 2a,b). The formation of numerous CFU-F colonies from the CMSCs samples after the 1st passage was also observed.

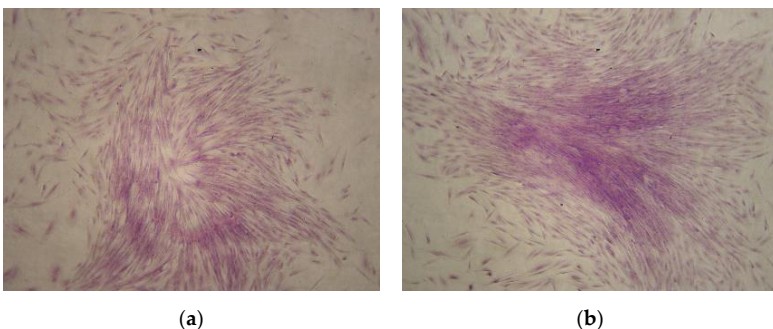

(a)                                            (b)

**Figure 2.** (**a**,**b**) Colony-forming unit fibroblast (CFU-F) colonies formation from CMSCs after passage 1 with Romanowsky–Giemsa staining (10× magnification of objective).

## 3.3. Characterization of CMSCs

In vitro cultured CMSCs were characterized by flow cytometry. The panel of markers was adapted to the criteria of the International Society for Cell Therapy (ISCT) [10]. CMSCs isolated from human chorion, after passage 1, showed expression of characteristic mesenchyme expression and absence of hematopoietic features. The cell–surface markers analyzed using FACS revealed that CMSCs after passage 1 had positive expression of CD105, CD90, CD73, CD29, CD44, and CD54 and negative expression of CD45 and CD34. CMSCs cells were positive for many markers common to MSCs. The graph in Figure 3a,b shows the expression of the markers observed. Multilineage CMSCs potential was tested for the three following types—osteogenic, chondrogenic, and adipogenic differentiation. After 2–3 weeks, the presence of calcium deposits, chondrogenic matrix, and lipid vacuoles was observed. The differentiating ability of CMSCs under suitable stimulatory culture conditions was confirmed (data not shown).

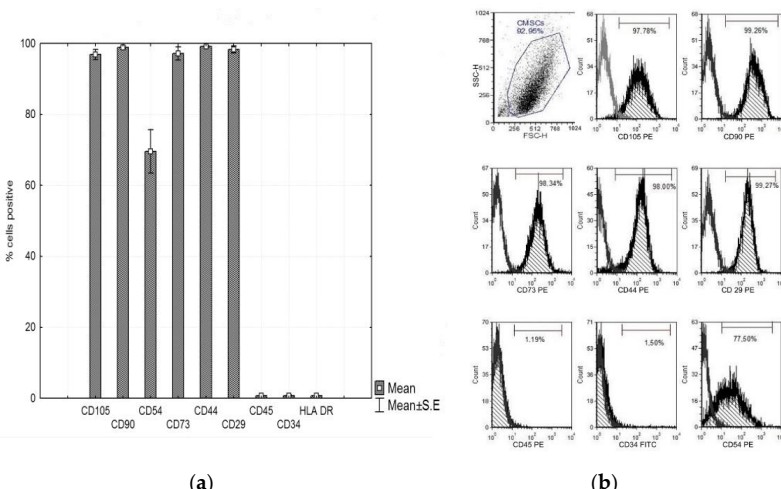

(a)                                            (b)

**Figure 3.** (**a**) Phenotype expression of mesenchymal and haematopoietic markers of CMSCs at passage 1, CD105+, CD90+, CD73+, CD29+, CD44+, CD54+, CD45−, and CD34−. Histograms represent mean ± S.E of seven independent experiments and (**b**) representative flow cytometry analysis for CMSCs at passage 1.

### 3.4. Characterization of Lymphocytes

CFSE-labeled PHA lymphocytes were cultured with CMSCs for 4 days. After culturing, the lymphocytes were analyzed by flow cytometry. The marker panel was CD4, CD193, CD3, CD69, and CD119. The lymphocytes then had positive expression of CD4 and CD193 and negative expression of CD3, CD69 and CD119. The graph shows the expression of the observed markers (see Figure 4a,b).

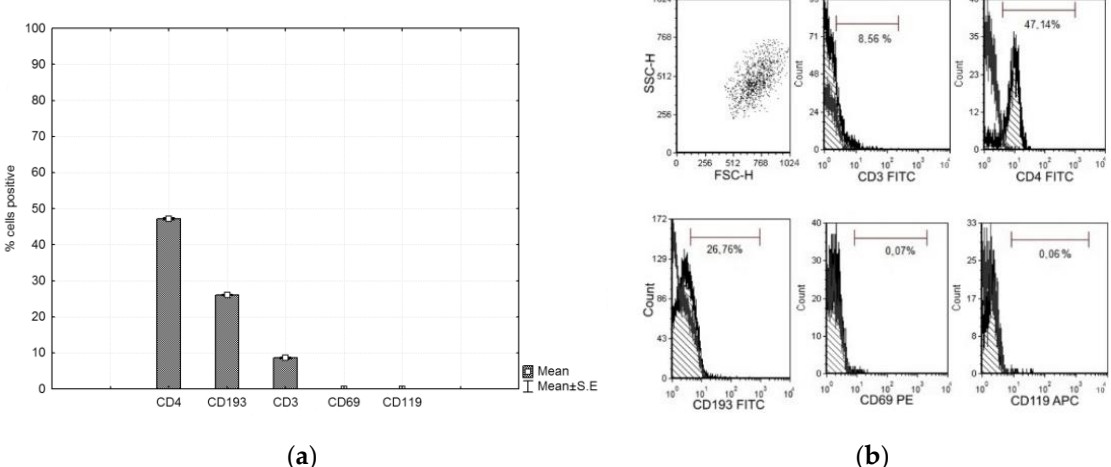

(**a**)                                                (**b**)

**Figure 4.** (**a**) Phenotype expression of markers of the phytohemagglutinin (PHA) lymphocyte. Histograms represent mean ± S.E of seven independent experiments and (**b**) representative flow cytometry analysis for lymphocytes.

### 3.5. Impact of CMSCs on Lymphocytes Proliferation

The immunomodulatory effect of CMSCs on activated PHA lymphocytes that were fluorescently labelled with CFSE was studied. CMSCs interacted with lymphocytes in co-culture in vitro (see Figure 5a,b). Lymphocytes proliferation assays are based on the ability of lymphocytes to proliferative when activated. CFSE has been widely used for investigating lymphocyte division and mitotic activity [34]. Carboxyfluorescein diacetate succinimidyl ester passively diffuses into cells. Each new generation in a population of proliferating lymphocytes is marked by twofold decrements in cellular fluorescence intensity, which can be detected by flow cytometry. Proliferation of three types of groups of CFSE labeled PHA lymphocytes were compared. Groups of lymphocytes and co-cultured with CMSCs *Poly (I:C)*, lymphocytes with CMSCs without stimulation and lymphocytes alone were compared. The inhibitory effect of CMSCs was tested after 4 days of co-culture with PHA lymphocytes. CMSCs were co-cultured at a ratio (1:4) to activated lymphocytes. The ability to influence proliferation was tested in the PHA lymphocyte group. The impact on proliferation was recorded by measuring the change in CFSE fluorescence intensity (see Figure 6).

A suppressive effect on lymphocyte proliferation was observed after 4 days of co-culture of the following groups of cells. CMSCs were able inhibit the proliferation of activated lymphocyte independently of *Poly (I:C)* activation. Statistically significant differences were detected between experimental groups lymphocytes (LY), groups: LY/CMSCs *Poly (I:C)* and LY/CMSCs against the group of lymphocytes cultured alone. Post hoc analysis confirmed the statistical differences in significance level between the compared group of lymphocytes co-cultured LY/CMSC *Poly (I:C)* ($p$ = 0.0028) and the lymphocytes LY/CMSC ($p$ = 0.0243). In both groups the percentage difference in proliferation was statistically significant and confirmed by the test (ANOVA, $p$ = 0.0052) (see Figure 7a). No statistical significance was confirmed between lymphocytes and CMSCs *Poly (I:C)* and lymphocytes with CMSCs ($p$ = 0.6645). The CMSCs groups and the group lymphocytes alone cultured. The Δ proliferating fraction (ΔPF) was calculated by subtracting the mean background proliferation from the mean proliferating

fraction in response to stimulation: $\Delta PF_1$ = [%CFSE Ly CMSCs *Poly (I:C)*] − [%CFSE Ly] and $\Delta PF_2$ = [%CFSE Ly(CMSCs)] − [%CFSE Ly]. The stimulation index (SI) was calculated as the percentage of Ly CMSCs *Poly (I:C)* divided by the percentage Ly or calculated as the percentage of Ly (CMSCs) divided by the percentage Ly. Change in proliferation was marked as significant at both the levels $\Delta PF \geq 1\%$ and SI $\geq 3.0$. Measurements found the mean $\Delta PF1 = 29.61 \pm 18.93\%$ and the mean $\Delta PF2 = 25.75 \pm 18.49\%$. Paired t-test showed that statistically there was no significant difference between $\Delta PF1$ and $\Delta PF2$ ($p = 0.1380$) (see Figure 7b).

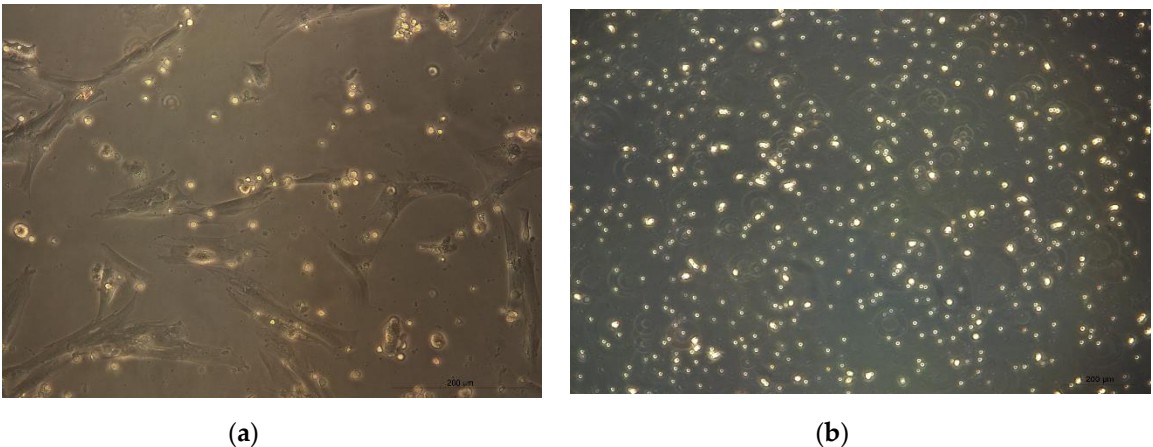

(**a**)　　　　　　　　　　　　　　　　　　　　　　　　(**b**)

**Figure 5.** (**a**) CMSCs interact with lymphocytes in co-culture in vitro. CMSCs are adherent spindle shapes and lymphocytes have round shape (20× magnification of objective). (**b**) Activated PHA lymphocytes cultured separately (20× magnification of objective).

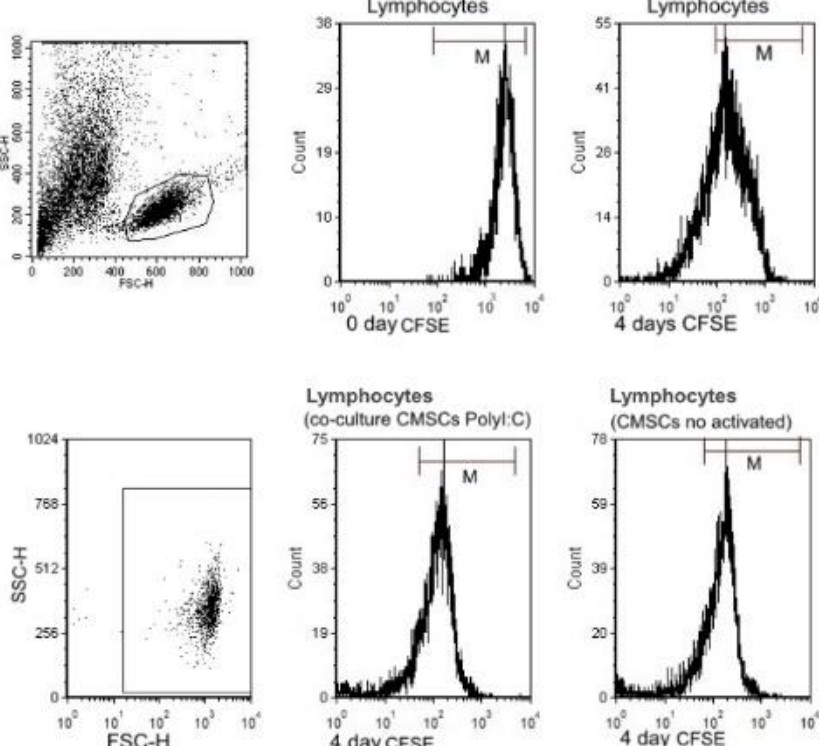

**Figure 6.** Flow cytometry analysis of carboxylflurescenin succinidyl ester (CFSE) fluorescence intensity of three groups of co-cultured PHA lymphocytes CFSE and CMSCs. Inhibitory effect of CMSCs after 4 days of co-culture. Representative flow cytometry analysis for lymphocytes.

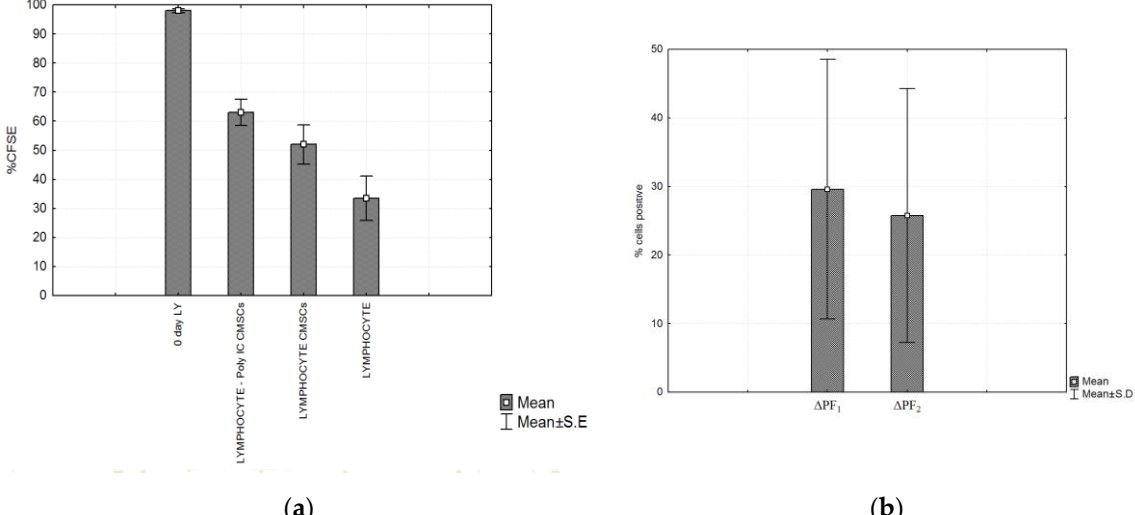

**Figure 7.** (**a**) Compared proliferation of three types of groups of CFSE labeled PHA lymphocytes. The highest fluorescence was observed on the first day (0 day lymphocytes (LY)). After 4 days the fluorescence was compared in the three experimental groups. Significant differences between the groups: LY/CMSCs *Poly (I:C),* LY/CMSCs and group lymphocytes alone cultured was detected. LY/CMSC *Poly (I:C)* ($p = 0.0028$) and lymphocytes LY/CMSC ($p = 0.0243$) and (**b**) Post hoc analysis of the experimental group of lymphocytes, CMSCs (*Poly I:C*) ($p = 0.0028$) and lymphocytes with CMSCs ($p = 0.0243$). Comparison of the Δ proliferating fraction (ΔPF) of groups ΔPF1 CMSCs *Poly (I:C)* group and ΔPF2 (CMSCs). Histograms represent mean ± S.E of seven independent experiments.

### 3.6. Impact of MSCs on Expression of Markers Involved in Modulating the Immune Response

After contact co-culture of CMSCs and activated lymphocytes, the expression of markers involved in the modulation of the immune response was evaluated. The expression of the monitored markers and the intensity of CFSE fluorescence were determined by flow cytometry on a FACS Calibur instrument (Becton Dickinson) using Cellquest software (Becton Dickinson) and FSC Express 5 (De Novo software). The expression of CD90, CD44, CD54, IL-10, TLR4 and TLR3 on CMSCs was evaluated in the presence of lymphocyte. The change of expression was monitored in activated CMSCs after stimulation of *Poly (I:C)* and co-cultivation with activated PHA lymphocytes and non-activated CMSCs. Surface molecules like chemokine receptors and adhesion molecules mediated direct cell-to-cell communication. The expression of CD90, CD44. CD54, IL-10 and TLR3 and TLR4 receptor expression was followed. CD90 thymocyte differentiation antigen 1 (Thy-1) is the cell surface specific marker for mesenchymal stem cells. CD90 was expressed in most of the cell populations studied. CD44, a cell surface receptor, is considered to be one of the most important adhesion molecules, responsible for activated T-cell mobilizing into the process of inflammation. The expression dynamics of marker CD44 were also monitored. It is a receptor for hyaluronic acid (HA), which is involved in intercellular interactions as well as reactions with collagen molecules. CD44-HA binding facilitates MSC migration and adhesion. Marker CD44 was increased and positive in the *Poly (I:C)* CMSCs group compared to the observed group of inactivated CMSCs. In the group of inactivated CMSCs, the marker CD44 had a significantly lower expression. CD54 (ICAM-1) is highest expressed molecules on MSCs. ICAM-1 is a transmembrane protein that belongs to the immunoglobulins superfamily [35]. Inflammatory cytokines can mediate the interaction of ICAM-1 and LFA-1 ligand and subsequently promote leukocyte migration to the site of tissue or organ damage. Activated CMSCs had significantly higher CD54 expression compared to the lower CD54 expression of non-activated cells. *Poly (I:C)* acts as a TLR3 activator on cells, an analogue of double-stranded viral RNA. MSCs react dynamically to molecules of the surrounding microenvironment and, after activation, acquire strong immunomodulatory properties. Based on these observations, a type of polarization of MSCs on MSC1 and MSC2 associated with TLR4 activation, or TLR3, was proposed.

MSC1 with a pro-inflammatory phenotype and activated TLR4 are characterized by the secretion of inflammatory mediators, IL-6 and IL-8. TLR3-activated MSC2s produce the anti-inflammatory mediators IL-10 and IL-1RA, which can inhibit T cell proliferation through prostaglandin E2 (PGE2) and IDO expression [36]. MSCs that are active through TLR3 have a more effective ability to activate their adhesive properties and bind to leukocytes. Low expression of TLR3 and TLR4 was observed on the CMSCs after *Poly (I:C)* activation and in the group without activation. MSCs produce IL-10, which acts as a broad-spectrum, anti-inflammatory cytokine. Effects on IL-1 function. It suppresses the production of tumor necrosis factor beta (TNF-β) and other pro-inflammatory factors. In the *Poly (I:C)* treated group cells, increased IL-10 expression after co-cultivation was observed, compared to CMSCs that were not activated. The overall results of the CMSCs phenotype after co-cultivation with lymphocytes are as follows. *Poly (I:C)* CMSCs stimulated had positive expression of CD90, CD44 and CD54, less positive IL-10 and negative of TLR4, and TLR3. Non-stimulated CMSCs had positive expression of CD90, less positive of CD44 and negative of CD54, TLR4, TLR3, and IL-10. The difference in the sample of activated *Poly (I:C)* CMSCs was mainly the expression in the markers CD54 and IL-10.

The expression of TLR3 and TLR4 receptors, which are associated with the pathogens-associated molecular pattern (PAMP) and damage-associated molecular pattern (DAMP) was monitored. Upon TLR activation, CMSCs polarize and trigger several intracellular cascades that lead to the production and release of multiple cytokines. The change in expression in activated CMSCs was observed after the fourth passage, after *Poly (I:C)* stimulation and co-culture with active PHA lymphocytes. The phenotype and expression of the markers in the CMSCs sample was determined. The expression of the marker CD90 was also monitored. CD90 is a hallmark of MSCs and is involved in cell adhesion and migration. We also monitored the expression dynamics of the marker CD44, a receptor for hyaluronic acid, that also participates in intercellular interactions and interactions with collagen molecules. The dynamics of CD54 expression was monitored after co-culture with lymphocytes. CD54 is an immunoglobulin group adhesive molecule. It is a transmembrane protein with an extracellular domain. CD54 is a ligand for immune-associated ligands. CD54 is a binding site for antigen associated with leukocyte function (LFA-1). The expression of IL-10 was monitored. It acts as an anti-inflammatory inhibitor of pro-inflammatory cytokines such as TNF-alpha (see Figure 8a,b).

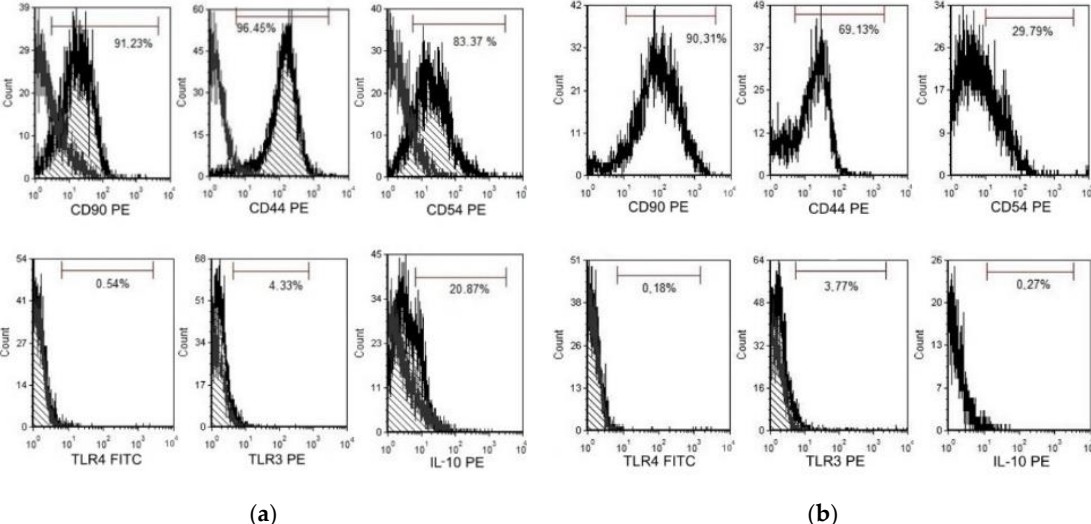

(**a**)                    (**b**)

**Figure 8.** (**a**) Flow cytometry (FACS) analysis chorionic mesenchymal stem cells activated with polyinosine: polycytidylic acid CMSCs *Poly (I:C)* co-culture with lymphocyte and (**b**) FACS analysis CMSCs non-activated CMSCs co-culture with lymphocyte. Representative flow cytometry analysis.

In comparison, activated non-activated CMSCs by paired t-test showed statistically significant differences in the groups CD54 (%) ($p = 0.0000$), CD44 (%) ($p = 0.0006$) and IL-10 (%) ($p = 0.0000$); differences were not confirmed in the CD90 (%) ($p = 0.8662$), TLR3 (%) ($p = 0.3752$) and TLR4 (%) ($p = 0.0925$) groups (see Figure 9 and Table 1).

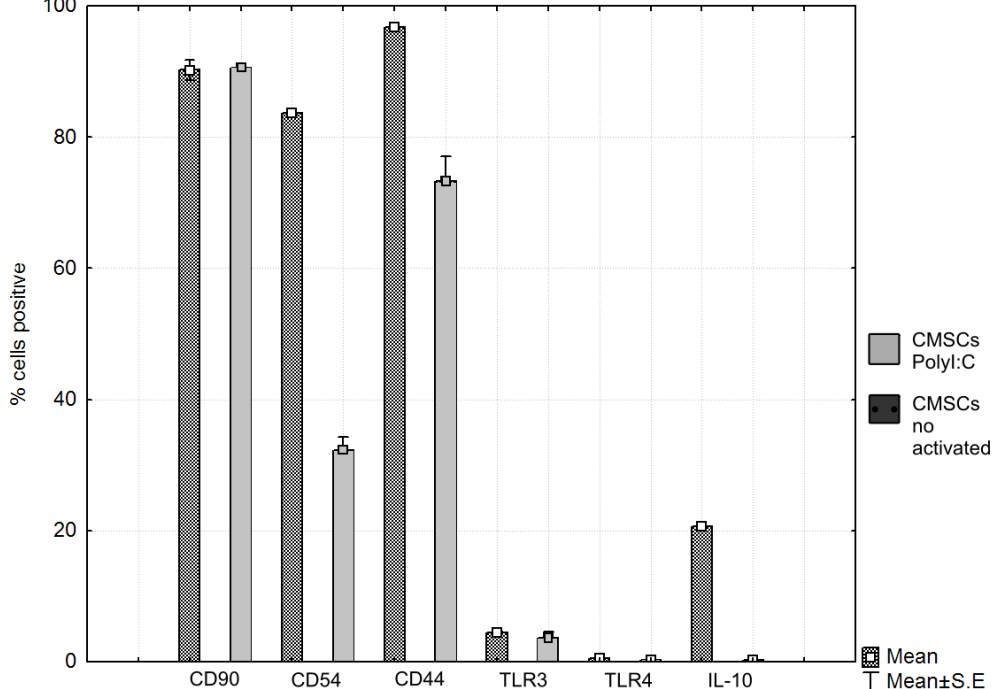

**Figure 9.** Phenotype expression markers co-cultured chorionic mesenchymal stem cells activated with polyinosine: polycytidylic acid CMSCs *Poly (I:C)* and CMSCs by paired t-test showed differences in the groups CD54 (%) ($p = 0.0000$), CD44 (%) ($p = 0.0006$), and IL-10 (%) ($p = 0.0000$); differences were not confirmed in the CD90 (%) ($p = 0.8662$), TLR3 (%) ($p = 0.375200$) and TLR4 (%) ($p = 0.0925$) groups, t-test. Histograms represent mean ± S.E of seven independent experiments. TLR = Toll-like receptor.

**Table 1.** Significant differences of CD44, CD54, and IL-10 between phenotype markers in co-cultured CMSC *Poly (I:C)* and CMSC.

| Phenotype Expression of Markers [%] | CD90 [%] | CD44 [%] | CD54 [%] | TLR4 [%] | TLR3 [%] | IL-10 [%] |
|---|---|---|---|---|---|---|
| CMSCs *Poly (I:C)* | 91.2 | 96.4 | 83.3 | 0.5 | 4.3 | 20.8 |
| CMSCs | 90.3 | 69.1 | 29.7 | 0.18 | 3.7 | 0.2 |

### 3.7. Statistical Analysis

All statistical tests were conducted using Statistica 9 software for multiple comparison of more than three experimental groups. ANOVA analysis with the Tukey post hoc test was performed, while t-tests were used for comparisons between two groups. Statistical significance occurred for a *p*-value less than 0.05 (typically ≤ 0.05). Data are presented as mean percentages of tested groups CMSCs ± standard error.

## 4. Discussion

Adult stem cells are widely used in regenerative medicine for their multiline and regenerative potential [37]. The study of the properties of mesenchymal stem cells is very important for their properties and their perspective use in regenerative medicine. In this work, we discuss the immunomodulatory properties of MSCs and the effects on lymphocytes proliferation on MSCs. The most commonly used source of mesenchymal stem cells is bone marrow. Later sources of

MSCs were expanded by other tissues, especially fatty tissue. The authors observed a significant reduction in interferon (IFN) gamma production of activated T cells after co-cultivation with human exosomes adipose MSCs fatty tissue [38]. Fetal membrane is a promising source of mesenchymal cells. Chorion was used as a source of MCSs. Unlike embryonic stem cells, chorionic mesenchymal cells can be experimentally used without ethical conflict, in contrast to embryonic stem cells. Sources of perinatal stem cells include umbilical cord blood, the Wharton's Jelly matrix of umbilical cord, amniotic fluid, the amnion and chorion. Perinatal tissues are important sources of stem cells. Perinatal stem cell is a type of cell that partially has the properties of postnatal stem cells and the properties of pluripotent, embryonic stem cells. Chorionic MSCs expressed genes associated with the undifferentiated cells NANOG, OCT4, and REX1 [39]. Due to the narrow ontogenetic relationships with embryonic stem cells, perinatal stem cells have lower immunogenicity. They contain a higher proportion of naive, spontaneous T lymphocytes, a small percentage of NK cells and overall lower immunogenicity. Differences between individual MSCs were observed [40]. The distinct difference in mesenchymal cells from individual sources is their different proliferative activity. Chorionic MSCs have higher proliferative activity compared to bone marrow and adipose tissue MSCs. Lower proliferation potential was recorded only in MSCs isolated from the amniotic membrane [41]. Placenta-derived cells have multilineage differentiation potential similar to MSCs in terms of morphology, cell-surface antigen expression, and gene expression patterns [42]. The immunomodulatory properties of CMSCs are less explored. The study focuses mainly on the effect of CMSCs on lymphocytes as well as monitoring the proliferation of activated lymphocytes via CMSCs. Characteristic features of mesenchymal stem cells isolated from the chorion, phenotype, adherence, and formation of CFU-F were observed. Expression of the major features of CD105+, CD90+, CD73+, CD44+, CD29+, CD54+, CD45−, and CD34− hematopoietic stem cells were also observed. Difference in the expression of these characters depending on the length of cultivation was detected. In the CMSCs primo-culture, the expression of all the mesenchyme features was detected at a lower percentage. The presence of hematopoietic markers was partially observed. In the following passages, the expression of hematopoietic markers was not detected. Monitored cells populations of the CMSCs retained expression of the mesenchymal traits in all seven isolated CMSCs samples between the first and third passage of observation. The preservation MSCs phenotype was confirmed while testing CMSCs immunomodulatory properties.

The main purpose of the study was to test the effect of CMSCs on a selected population of immune-active cells. Interactions between cells require intercellular contact and multiple solubility factors. Both cell populations, were monitored in co-cultivation. CMSCs were stimulated with *Poly (I:C)*, similar to viral DNA acting as a TLR3 ligand, in order to activate CMSCs during co-cultivation with activated lymphocytes. A co-cultivation system was used in vitro in order to assess the effect of activated vital fluorescence-labelled CFDA-SE lymphocytes and CMSCs. The intensity of fluorescence in co-cultivation was compared with the pure culture of activated and labelled lymphocytes. Lymphocytes phenotype and CMSCs phenotype were also evaluated after co-cultivation. Results from the following groups were assessed: CMSCs activated *Poly (I:C)* and CMSCs not activated. In co-cultivation of lymphocytes CMSCs reduced proliferation of activated lymphocytes was observed. The extent of proliferation decline was determined by the fluorescence difference of CFDA-SE labelled lymphocytes over four days of co-cultivation of lymphocytes and CMSCs. In the comparison groups, the co-cultivation of CMSCs and lymphocytes and the second group of isolated lymphocytes, the difference in proliferation of activated lymphocytes was evaluated. The suppression of proliferation of activated lymphocytes stained with CFSE in co-cultivation with CMSCs was observed. The co-cultured lymphocytes fluoresced to a greater extent than the activated self-cultured lymphocytes, which exhibited decreased fluorescence and thus underwent multiple mitoses. In this group, we saw inhibition of activated lymphocytes proliferation. This fact confirmed the immunosuppressive effect of CMSCs indirect contact culture on activated lymphocytes. Suppression proliferation of activated lymphocytes was noticed. Results indicate that CMSCs have immunomodulatory capacity.

MSCs respond to the changing environment. In the inflammatory environment, activation of MSCs occurs by occupying specific TLR3 and TLR4 receptors that are associated with the DAMP and PAMP pathogens. This stimulation is alternatively possible via the synthetic analogue of double-stranded (ds) RNA and *Poly (I:C)*. *Poly (I:C)* was used to stimulate CMSCs. Afterwards, the expression of surface and adhesion markers was tested. MSCs have enhanced expression of ICAM-1, the vascular cell adhesion molecule 1 (VCAM-1), and chemokine ligand 3 (CXCR3) ligands that are CXCR3-activated T lymphocyte ligands under inflammatory conditions. Many processes in the immune environment are caused by interactions of adhesion molecules. The molecules mediate the accumulation of immune cells and allow their close contact with MSCs. MSCs come into contact with T cells in the inflammatory environment. It is assumed that ICAM-1 and VCAM-1 have a significant effect on the MSCs-mediated immunomodulation. The expression of the adhesion molecules MSCs increases with inflammatory cytokines. Interactions between VCAM-1 and endothelial integrins have been demonstrated. VCAM-1 is a late-activating antigen ligand 4 (VLA-4) and ICAM-1 is a ligand for LFA-1. Recent studies have shown that after stimulation, MSCs produce inflammatory cytokines that are attractants for lymphocytes. The ICAM-1 and VCAM-1 molecules are considered to be co-stimulatory in the immune response. ICAM-1 and VCAM-1 play an important role in MSCs-mediated immunosuppression, thanks to their ability to interact with lymphocytes and MSCs. These adhesion molecules play a crucial role in specific immune responses to foreign pathogens. Adhesive molecules interact with lymphocytes and endothelium. Upon reaching the endothelium, adhesion molecules mediate cell movement [43]. Intercellular adhesion by ICAM-1 and VCAM-1 is essential for the activation of T-lymphocytes during the process of inflammation. An interaction between ICAM-1 receptor MSCs and the LFA-1 receptor on lymphocytes was observed. It plays an important role in triggering an immune response. MSCs co-cultivated with T lymphocytes in the presence of activation of the T lymphocyte antigen receptor significantly enhance the T lymphocyte adhesion capability by increasing the expression of ICAM-1 and VCAM-1. ICAM-1 and VCAM-1 are inducible in the presence of IFN-$\gamma$ and tumor necrosis factor alpha (TNF-$\alpha$) and IL-1. The immunosuppressive effect of MSCs on T lymphocytes was compared with the change in the intensity of adhesion molecule expression. Higher ICAM-1 expression in activated CMSCs compared to non-activated CMSCs was found. Authors [44] noticed increased expression of ICAM-1 and VCAM-1 after activated splenocytes co-stimulated with activated splenocytes. Adhesive molecules have a significant effect on motility functions. Activated splenocytes were able to enhance the expression of ICAM-1 and VCAM-1. Intercellular contact between lymphocytes and MSCs is required to increase ICAM-1 and VAM-1 levels. It is assumed that increased expression of ICAM-1 and VCAM-1 influences the immunomodulatory ability of MSCs [45]. The stimulation of CMSCs was observed and the presence of IL-10 was also noticed after co-cultivation of CMSCs and lymphocytes. The levels of IL-10 were higher in co-cultivated CMSCs, and the results were statistically significant.

The authors [46] observed multiple tissue sources of MSCs. Results show that Wharton's MSCs cause a weak immune response. On the other hand, they have the most powerful immunosuppressive potential compared to MSCs from bone marrow. Recent studies have shown that MSCs respond to danger signals in the environment of inflammation. MSCs express Toll-like receptors in which activation results in significant change and expression of the MSCs and polarization of MSCs on MSC1 or MSC2 phenotypes by TLR4 or TLR3 stimulation. MSC1 has a proinflammatory phenotype and secretion of inflammatory IL-6 and IL-8 interleukins. MSC2 produces anti-inflammatory interleukins [47].

Recent studies have confirmed that the immunomodulatory function of MSCs is coordinated with several types of immune cells. The detailed course of these interactions remains unclear. Several mechanisms of interaction of MSCs and immune cells have been described. In the intercellular contact of MSCs and immune system cells, activation secretion is triggered and in some cases immune cell apoptosis can be induced. It can be said that MSCs are a specific immune cell function coordinator in a particular setting. Immune systems and cells associated with the immune system are linked by processes that result in the production of cytokines and the change in phenotypic traits. Interactions can act to induce inflammatory processes. Many studies report conflicting results in monitoring the

production of cytokines and their effects. Not much is known about the function of the cells, which act by immunomodulation. There is some data on the impact of MSCs that are directly active in immunomodulation. MSCs can act suppressively on T lymphocytes, B lymphocytes, antigen-presenting cells (APC) and natural killers (NK). MSCs further act on the differentiation of immune cells and influence the formation of an immunoregulatory phenotype of immune cells with tolerogenic properties. MSCs enter into interactions with regulatory T lymphocytes, regulatory B lymphocytes, regulatory APCs and NK cells, and act together to create a tolerogenic environment that modulates the immune response [48,49].

## 5. Conclusions

Characteristic properties of mesenchymal stem cells isolated from chorion were confirmed along with phenotype, adherence to a plastic culture surface, fibroblast-like bipolar shape cells morphology and clonogenic potential. It was found that CMSCs interact with lymphocytes in co-cultivation. MSCs were demonstrated to be poor stimulators of an in vitro allogenic T cells response and they fail to induce activation of allogenic T cells. MSCs were shown to suppress both naive and memory T lymphocyte activation and proliferation induced by alloantigens and mitogens [50]. Results indicate that that fetal MSCs affected the proliferation of activated lymphocytes. The immunomodulatory ability of CMSCs was compared and tested in activated lymphocyte co-cultivation. The proliferation of activated lymphocytes was suppressed. The extent of proliferation decline was determined by the fluorescence difference of CFDA-SE-labeled lymphocytes over four days of co-cultivation of lymphocytes and the CMSCs. During the co-cultivation, changes in the phenotype of CMSCs CD44 and CD54 adhesion molecules and IL-10 were observed. Higher expression of adhesion molecules was detected after co-cultivation with activated lymphocytes. Results show that CMSCs act as immunosuppressives on activated lymphocytes, with phenotype change and increase of adhesion molecules allowing direct intercellular contact of interacting cell populations. MSCs are positively capable of acting immunomodulatory through the release of bioactive molecules as well as through direct cell contact [51]. Overall, the ability of MSCs with immunomodulatory properties was observed to contribute to immune system homeostasis and balance in immunological activities. To conclude, CMSCs appear to be a highly promising source of multipotent mesenchymal cells in experiments and therapeutic use.

**Author Contributions:** Conceptualization, data curation, supervision, and funding acquisition, J.Ž. and R.H.; methodology, validation, and project administration, M.T.; software, L.Z.; visualization, G.I.; formal analysis and investigation and writing—original draft preparation, D.B.; formal analysis, K.Š. All authors have read and agreed to the published version of the manuscript.

**Funding:** This research was funded by the Slovak Agency for Research and Development, grant number APVV-17-0278 and the Cultural Educational Grant Agency of the Ministry of Education, Science, Research and Sport of the Slovak Republic, grant number KEGA 023TUKE-4/2020, KEGA 041TUKE-4/2019, IMTS 26220220185, ITMS2014+ 313011D103 and incentives for research and development Stimuly BE 81 3DP TISSUE.

**Acknowledgments:** The authors thank Ján Rosocha, Head of the Associated Tissue Bank, University Hospital of L. Pasteur Košice for professional help and support within the project APVV-17-0118.

**Conflicts of Interest:** The authors declare no conflict of interest.

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
