# Peer review of "Analysis of Same Selected Immunomodulatory Properties of Chorionic Mesenchymal Stem Cells"

_applsci, doi:10.3390/app10249040_

Round 1

Reviewer 1 Report

In this study, the author demonstrated the immunomodulatory properties of CHorionic MSCs. However, this has been explored before in the literature although the novelty in this study is the expression of IL10 in the same time there was no other source compared to CMSCs such as Bone Marrows MSCs.  

Minor correction to consider:

1- the lymphocyte donors were they multiple individuals or one donor?

2-in Figure 5 I Would recommend to include picture of activated lymphocyte only so the reader can see the difference. 

3- Figure 6 the histogram of the lymphocyte after 4 days is not showing clearly the proliferation of lymphocyte 

Author Response

Applied Sciences

Applied Biosciences and Bioengineering                                                        8th December 2020

Manuscript ID: applsci-1033591

Dear Reviewer:

Thank you very much for giving us the opportunity to improve and resubmit our manuscript:

 Authors of the article, Analysis of same selected immunomodulatory properties of chorionic mesenchymal stem cells, are as follows - Darina Bačenkova, Marianna Trebuňová, Lukáš Zachar, Radovan Hudák, Gabriela Ižaríková, Katarína Šurínová, Jozef Živčák. Enclosed please find the revised manuscript.

Based on the comments raised by the reviewer, the manuscript was revised to the best of our ability and knowledge. Changes in the manuscript are underscored through "Track Changes" function in
Microsoft Word.

We would also like to thank the reviewer for the constructive and competent criticism. We believe that

our manuscript will be acceptable for publication.

Response to Reviewer:

  1. The lymphocyte donors - were they multiple individuals or one donor?

Answer: Venous blood samples were used from seven donors in the experiment.

  1. In Figure 5, I would recommend to include picture of activated lymphocyte only, so that the reader can see the difference. 

Answer: Figure 5 (B) - the picture of CFSE lymphocytes was, based on the reviewer’s recommendation, replaced with the picture of activated lymphocytes.

  1. Figure 6 - the histogram of lymphocytes after 4 days of cultivation is not clearly showing the proliferation of lymphocytes

Answer:  Figure 5 (B) – based on the reviewer’s recommendation, histogram of lymphocytes was used after 4 days of cultivation. Line 294 - the image of lymphocytes after 4 days of cultivation was used.

Sincerely, Darina Bačenková

Technical University of Košice, Faculty of Mechanical Engineering, Department of Biomedical Engineering and Measurement, Letná 9, 042 00, Košice, Tel.: +42 1055 602 2380

Reviewer 2 Report

Bačenkova et al present the article “Analysis of same selected immunomodulatory properties of chorionic mesenchymal stem cells” submitted for review to MDPI Applied Sciences. This is a relatively straightforward article with well-designed experiments and analysis complemented with an adequate write up analysing the CMSCs characterisation. There are however some minor corrections that need to be completed to bring this manuscript up to a publishable quality. There are several typographical and formatting errors that are easily fixable. There are a few item discrepancies below that the authors are recommended to re-check and correct before submitting the final version. I look forward to seeing this manuscript completed and published. Best wishes.

Line 85 and 92: “CD4+CD25+FoxP3+ Treg lymphocytes” needs punctuation as CD4+, CD25+, FoxP3+, Treg lymphocytes

Materials and methods:

Line 102. ethical guidelines: please provide the ethics approval number from the committee and date of approval, this is a standard requirement of MDPI journals.

Line 177. “FACS fenotyping of lymphocyte” should be “FACS phenotyping of lymphocytes”

Line 195.  “Co-cultivation CMSC and lymphocyte” should not be in bold font

Line 211. Kruskal Wallis should be hyphenated as Kruskal-Wallis

Line 217-219. should not be in bold font

Line 236. Please check the magnification these cells look to be photographed at 20x not 200x objective

Line 247. Please check the magnification these cells look to be photographed at 10x not 100x objective

Line 248. Title should not be in bold

Line 266-271. Should not be in bold

Line 278. Please check the magnification these cells look to be imaged at 20x not 200x objective

Line 282. Should not be in bold

Line 384. (and 389/390). Delete empty lines

Line 428. “Extraembryonic MSCs also have greater multipotent potential, than MSCs isolated from bone marrow or adipose tissue [42].”

Please amend the above sentence as Fukuchi et al did not make this claim, it was stated in their paper that “placenta‐derived cells have multilineage differentiation potential similar to MSCs in terms, of morphology, cell‐surface antigen expression, and gene expression patterns”

Line 435. Define “primo-culture”?

Line 500. Define KD?

Line 506. “Recent studies have confirmed that the immunomodulatory function of MSCs heals several types of immune cells.” The use of the word “heals” in this sentence is incorrect, it is also unclear what is meant to be stated here. Maybe “coordinates” or “interacts with”

Author Response

Applied Sciences

Applied Biosciences and Bioengineering                                                        8th December 2020

Manuscript ID: applsci-1033591

Dear Reviewer,

Thank you very much for giving us the opportunity to improve and resubmit our manuscript:

Authors of the article, Analysis of same selected immunomodulatory properties of chorionic mesenchymal stem cells, are as follows - Darina Bačenkova, Marianna Trebuňová, Lukáš Zachar, Radovan Hudák, Gabriela Ižaríková, Katarína Šurínová, Jozef Živčák. Enclosed please find the revised manuscript.

Based on the comments raised by the reviewer, the manuscript was revised to the best of our ability and knowledge. Changes in the manuscript are underscored through "Track Changes" function in
Microsoft Word.

We would also like to thank the reviewer for the constructive and competent criticism. We believe that

our manuscript will be acceptable for publication.

The text is formatted after editing and the line number are changed.

Response to Reviewer

Original line

Line 85 and 92: “CD4+CD25+FoxP3+ Treg lymphocytes” I fixed punctuation as CD4+, CD25+, FoxP3+, Treg lymphocytes - I fixed it

Line 102. ethical guidelines: I fixed the ethics approval number from the committee and date of approval, this is a standard requirement of MDPI journals. We added: Funding: KEGA 041TUKE-4/2019

Line 177. “FACS fenotyping of lymphocyte” I fixed it “FACS phenotyping of lymphocytes”

Line 195.  in bold font “Co-cultivation CMSC and lymphocyte” I fixed it “Co-cultivation CMSCs and lymphocytes

Line 195.  “Co-cultivation CMSC and lymphocyte” I fixed it 2.8. “Co-cultivation CMSCs and lymphocytes

Line 211. Kruskal Wallis I fixed it Kruskal-Wallis

Line 217-219. I fixed it - not be in bold font

Line 236. I fixed the magnification at 20x objective not 200x objective magnification of objective 20x, (20x magnification of objective) 200x).

Line 247. I fixed the magnification at 10x objective not 100x objective magnification of objective 10x, (10x magnification of objective) 100x).

Line 248. I fixed it. Title is not be in bold

Line 266-271 I fixed it. Text is not bold text

Line 278. I fixed the magnification at 20x objective not 200x objective magnification of objective 20x, (20x magnification of objective) 200x).

Line 282. I fixed it. Text is not bold text.

Line 384. (and 389/390). I fixed it. Deleted empty lines

Line 428 I fixed it on: “Placenta‐derived cells have multilineage differentiation potential similar to MSCs in terms, of morphology, cell‐surface antigen expression, and gene expression patterns. [42]

Line 500. Define KD? I fixed it. This is a mistake in the Slovak language: KD - kostná dreň – English: bone marrow.

Line 506. I fixed it.  “Recent studies have confirmed that the immunomodulatory function of MSCs coordinates with several types of immune cells.”

Sincerely, Darina Bačenková

Technical University of Košice, Faculty of Mechanical Engineering, Department of Biomedical Engineering and Measurement, Letná 9, 042 00, Košice, Tel.: +42 1055 602 2380

Round 2

Reviewer 1 Report

Thank you for considering the changing